# TRAINING-FREE WATERMARKING FOR AUTOREGRESSIVE IMAGE GENERATION

## ABSTRACT

Invisible image watermarking can protect image ownership and prevent malicious misuse of visual generative models. However, existing generative watermarking methods are mainly designed for diffusion models while watermarking for autoregressive image generation models remains largely underexplored. Moreover, direct application of LLM watermarking solutions may impair image diversity and compromise the imperceptibility of watermarks. To address these challenges, we propose IndexMark, a training-free watermarking framework for autoregressive image generation models. IndexMark is inspired by the redundancy property of the codebook: replacing autoregressively generated indices with similar indices produces negligible visual differences. The core component in IndexMark is a simple yet effective match-then-replace method, which carefully selects watermark tokens from the codebook based on token similarity, and promotes the use of watermark tokens through token replacement, thereby embedding the watermark without affecting the image diversity and quality. Watermark verification is achieved by calculating the proportion of watermark tokens in generated images, with precision further improved by an optional Index Encoder. Furthermore, we introduce an auxiliary validation scheme to enhance robustness against cropping attacks. Experiments demonstrate that IndexMark achieves state-of-the-art performance in terms of image quality and verification accuracy, and exhibits robustness against various perturbations, including cropping, noises, Gaussian blur, random erasing, color jittering, random rotation, and JPEG compression. Code will be made public.

## 1 INTRODUCTION

With the remarkable success of Large Language Models (LLMs) (Vaswani et al., 2017; Brown et al., 2020; Zhang et al., 2022) in natural language processing, recent advancements have seen autoregressive image generation models, such as LlamaGen (Sun et al., 2024a) and VAR (Tian et al., 2024), demonstrating substantial potential in the domain of visual generation. These models leverage a Vector Quantization (VQ) tokenizer (Van Den Oord et al., 2017) to transform images into discrete tokens. Subsequently, they autoregressively predict the "next token" within a codebook to generate images. Notably, these models exhibit significant advantages in terms of both image quality and generation speed. Open-source autoregressive models enable broad creation of customized images. However, it also brings potential risks of model misuse (Brundage et al., 2018; Zohny et al., 2023; Vincent, 2020), such as fake news fabrication, ambiguous copyright attribution, and improper use of public figures' portraits. Amidst growing calls for government regulation and industry compliance (Kelly, 2023; Wiggers, 2023), model developers need to enhance image traceability to ensure accountability in legal liability determination, copyright protection, and content moderation.

Invisible watermarking (Huang et al., 2024; Al-Haj, 2007) provides a technical pathway for image traceability. This technology embeds imperceptible watermarks into images to help model developers achieve user-level attribution tracking of AI-generated content. Existing watermarking methods can be broadly categorized into two types: post-processing watermarks embedded after generation (Cox et al., 2007; Xia et al., 1998; Al-Haj, 2007), and generative watermarks integrated during the generation process (Yu et al., 2020; Fernandez et al., 2023). Since the former introduces additional inference and storage overhead, generative watermarks are generally more practical and hence more popular. However, current generative watermarking techniques primarily focus on diffusion models and lack exploration for the emerging autoregressive image generation models. Due to substantial architectural

differences between the two paradigms—diffusion models employ progressive denoising (Ho et al., 2020) whereas autoregressive models rely on sequential generation (Sun et al., 2024a)—current diffusion-based watermarking methods cannot be directly applied to autoregressive models.

Recently, some watermarking research for LLMs (Kirchenbauer et al., 2023; Mao et al., 2025) has provided approaches for watermarking AR models. These methods embed a watermark by altering the statistical distribution of generated indices through a sampling bias at inference time. However, when applied to AR image models, these methods suffer from the following drawbacks: **1) Reducing Image Diversity:** These methods embed the watermark by interfering with the model's generative process. This external intervention severely compromises the model's performance, leading to a reduction in the diversity of output content (Liu et al., 2024; Xu et al., 2025). **2) Lack of Imperceptibility:** The codebook size of LLMs typically exceeds 200k, whereas the codebook capacity for image models is only 1k-10k. Consequently, when LLM watermarking techniques are adapted to the domain of autoregressive image generation, the probability of assigning a large number of visually similar indices to the same biased list increases significantly. This suboptimal allocation strategy can lead to the omission of specific color patches in the generated image (for instance, if most blue-toned indices are concentrated in a single biased list, the model will struggle to generate images containing a sky, as shown in Figure 5), which severely compromises the watermark's imperceptibility.

We believe *leveraging the characteristics of the autoregressive image generation models is the key to solving the aforementioned issues*. Recent research in autoregressive image generation models has identified a notable *redundancy* issue in their codebooks (Hu et al., 2025; Guo et al., 2025): a large number of vectors are associated with different indices but highly similar to each other. This feature naturally leads to an elegant solution of watermarking that has minimal impact on the content of the image. Specifically, we divide the codebook into "red" and "green" groups by pairing similar indices. After generating the indices, we replace as many red indices as possible with their paired green indices (called watermark tokens), thus changing the distribution ratio of red-green indices in the final sequence to embed watermark information (see Figure 1). This type of watermark has four advantages. **1)** No biased sampling is required during inference, thereby effectively ensuring the diversity of generated images. **2)**

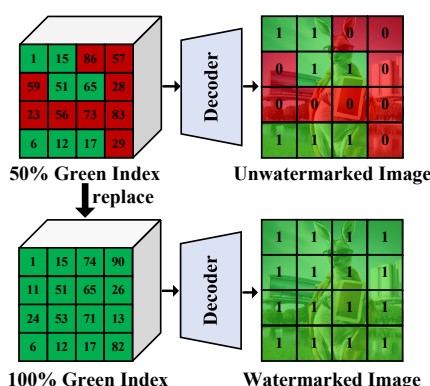

**50% Green Index**

**Unwatermarked Image**

replace

**100% Green Index**

**Watermarked Image**

Figure 1: Watermark embedding by index replacement to attain a higher proportion of watermark tokens (green index).

It is inherently robust and can only be removed by extensively modifying the color blocks of the image. **3)** The redundancy of the codebook allows this *match-then-replace* strategy to imperceptibly embed watermarks. **4)** Moreover, different red-green division schemes can correspond to different identity identifiers (IDs), assisting model developers in image tracing.

Building on the above insights, this paper proposes IndexMark, the first *training-free* watermarking method for autoregressive image generation models, which requires no retraining of the AR model. We first formulate the pairing of similar indices as a *maximum weight perfect matching* problem (Osiakwan & Akl, 1990), and solve it with top-K pruning and the Blossom algorithm (Edmonds, 1965). Then, we randomly assign each pair of indices to a red or green list. After autoregressive index generation, red indices are selectively replaced with their paired green indices according to index confidence, thereby embedding an invisible image watermark. This *match-then-replace* strategy robustly embeds watermarks with image quality and content well preserved. During the watermark verification stage, indices of the generated image are reconstructed via VQ-VAE to compute the "green-index rate" for verification. To compensate the index reconstruction errors of VQ-VAE, we introduce an Index Encoder for accurate index reconstruction. Although the red-green watermark is intrinsically robust against various image perturbations, the verifier is still vulnerable to cropping attacks due to VQ-VAE's block-level processing characteristics. Therefore, we propose a corresponding cropping-robust validation scheme specific to the modern AR image generation models.

Our key contributions are summarized as follows: 1) We propose a training-free watermarking framework that can be directly applied to autoregressive image generation models without requiring any additional fine-tuning or training. 2) We introduce a match-then-replace approach, which enables training-free watermark embedding with minimal impact on the visual quality of the images. 3) We

design a precise image indexing validation framework that can verify the presence of watermarks with higher statistical confidence. 4) Thanks to the Index Encoder and the designed cropping-robust watermark verification method, our approach demonstrates strong robustness towards a wide range of image perturbations.

## 2 RELATED WORK

### 2.1 IMAGE WATERMARKING

Watermarks in generative models can be embedded either after images are generated (*i.e.*, post-processing) or during the generation process (*i.e.*, in-processing). Post-processing methods are mainly divided into transformation-based and deep encoder-decoder methods. Representative post-processing methods include Discrete Wavelet Transform (DWT) (Xia et al., 1998), Discrete Cosine Transform (DCT) (Cox et al., 2007), and DWT-DCT methods (Al-Haj, 2007), which embed watermarks into the spatial or frequency domain. Deep encoder-decoder methods often generate watermarked images in an end-to-end manner (Zhang et al., 2019). However, these methods often struggle to generalize to images outside the training data distribution. Research on in-processing watermarking primarily focuses on diffusion-based models. The Tree-Ring watermark (Wen et al., 2023) embeds watermark into the noisy image before denoising. ROBIN (Huang et al., 2024) injects watermark into an intermediate diffusion state while maintaining consistency between the watermarked image and the generated image and robustness of the watermark. Though the performance is strong, these methods cannot be directly transferred to autoregressive architectures. Our proposed method requires no training and can be widely applied to codebook-based autoregressive models, effectively bridging the research gap in watermarking for autoregressive image generation architectures.

### 2.2 AUTOREGRESSIVE IMAGE GENERATION

In autoregressive image generation, image data is typically transformed into one-dimensional sequences of pixels or tokens, and the model predicts the next image token based on the existing context. Early autoregressive image generation models (Van den Oord et al., 2016; Van Den Oord et al., 2016) perform image generation by predicting continuous pixels, which have high computational complexity. The seminal work, VQ-VAE (Van Den Oord et al., 2017), builds a codebook containing feature representations and casts image generation into a discrete label prediction problem. VQ-GAN (Esser et al., 2021) extends VQ-VAE by using adversarial training to improve the image quality. Recently, LlamaGen (Sun et al., 2024b) and Open-MAGVIT2 (Luo et al., 2024) apply the concept of next-token prediction, which has been widely used in large language models, to autoregressive image generation, achieving performance that even surpass diffusion-based methods.[1] However, the problem of embedding watermarks into autoregressive image generation models remains largely understudied, exposing huge risks of model misuse for these models. While existing LLM watermarking approaches (Kirchenbauer et al., 2023; Mao et al., 2025) provide conceptual foundations for autoregressive models, their direct application compromises image diversity and generation quality. Our IndexMark method addresses these challenges through a match-then-replace approach, effectively bridging this research gap.

## 3 METHODOLOGY

**Our Framework** As illustrated in Figure 2, our IndexMark framework is composed of two parts: watermark embedding (Sec. 3.1) and watermark verification (Sec. 3.2). In the watermark embedding part, we first divide the codebook of an autoregressive model into pairs of indices such that each pair contains similar vectors. Then, for each index pair we randomly assign one index into a red list and the other into a green list. Finally, we perform selective red-green index replacement based on index confidence during the image decoding process to embed invisible watermarks into images (*i.e.*, replacing as many red indices as possible with green indices). In the watermark verification part, we propose a method based on statistical probability, where an Index Encoder is introduced to achieve precise reconstruction of indices. We also propose a cropping-invariant watermark verification scheme for cropped images.

---

[1]The background on autoregressive image generation can be found in the Appendix C.1.

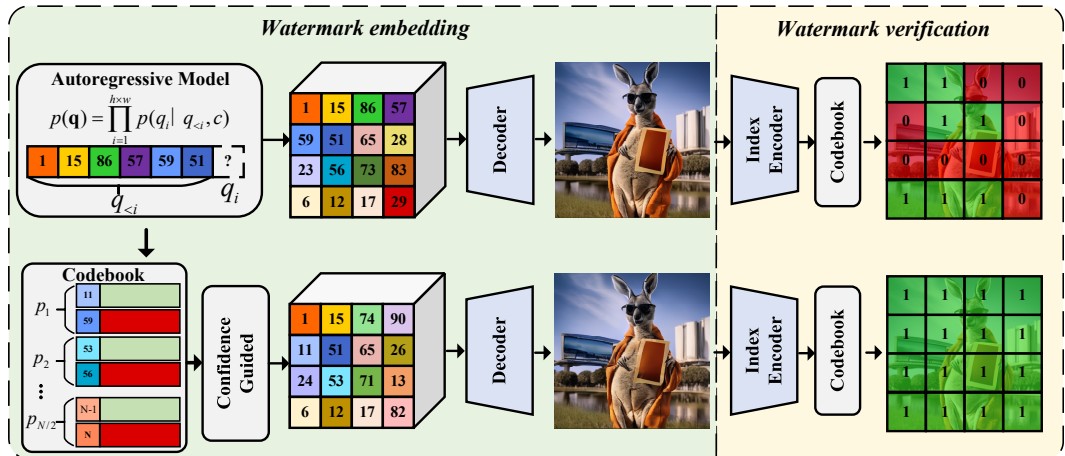

Figure 2: Watermark embedding and verification of IndexMark. During autoregressive index generation, IndexMark selectively replaces red indices with green indices from the same index pair based on confidence to embed the watermark. The watermarked image is fed into the Index Encoder to calculate the green index rate for watermark verification.

### 3.1 WATERMARK EMBEDDING

**Construction of Index Pairs** We aim to divide the indices in the codebook, $\mathcal{I} = \{1, 2, ..., N\}$, into index pairs $\mathcal{P} = \{p_1, p_2, ..., p_{N/2}\}$, where each pair $p_k = \{i_k, j_k\}$ contains two distinct indices from $\mathcal{I}$ such that $\bigcup_{k=1}^{N/2} p_k = \mathcal{I}$ and $p_k \cap p_{k'} = \varnothing$ for $k \neq k'$. The objective of the red-green index allocation is to compute an optimal assignment that maximizes the sum of the intra-pair similarity $S_{sum}$ for all red-green index pairs:

$$S_{sum} = \max_{\mathcal{P} \in \mathbb{P}} \sum_{k=1}^{N/2} \text{sim}(i_k, j_k), \tag{1}$$

where $\mathbb{P}$ is the set of all possible such partitions $\mathcal{P}$ and $\text{sim}(i, j)$ is the cosine similarity between the vectors of index $i$ and index $j$ in the codebook. We cast this problem into a *maximum weight perfect matching* problem. Specifically, we construct a complete graph $G = (V, E)$, where each vertex in the vertex set $V$ represents an index from the codebook and the edge set $E$ connects all pairs of vertices, with the edge weights $w$ set as the cosine similarity between the two connected vertices. After constructing this complete graph, our objective is to find a maximum weight perfect matching $M^*$, which is a subset of $E$ containing $N/2$ edges such that every vertex in $V$ is linked to only one edge in $M^*$, and the sum of the weights of these $N/2$ edges is maximized:

$$M^* = \arg\max_{M \in \mathbb{M}} \sum_{(i,j) \in M} w(i, j), \tag{2}$$

where $\mathbb{M}$ represents the set of all possible perfect matchings on the graph $G$, and $w(i, j)$ represents the weight of the edge connecting vertex $i$ and vertex $j$.

We solve this problem using the Blossom algorithm (Edmonds, 1965). Considering the large number of indices in the codebook, directly applying the Blossom algorithm would result in extremely high computational complexity. For this reason, we perform top-K pruning on the complete graph, retaining only the $K$ edges with the highest weights for each vertex. We then apply the Blossom algorithm (Edmonds, 1965) to the pruned sparse graph to obtain the maximum weight perfect matching $M^*$. Please refer to Appendix C.2 for the details on the Blossom algorithm.

**Red-Green Index Assignment** After obtaining the maximum weight perfect matching $M^*$, we need to assign indices to red and green lists for each index pair in $M^*$. For simplicity, we randomly assign the two indices in each index pair to the red and green lists. In practical applications, users can customize the assignment of red and green indices. The total number of possible assignments is as high as $2^{N/2}$, providing model developers and users with extremely abundant identity identifiers (IDs) for image tracing. We also provide an estimation of the watermarking system's capacity under the premise of avoiding user confusion in Appendix C.4.

**Confidence-Guided Index Replacement**    Autoregressive image generation models produce token index sequences in an autoregressive manner. Our objective is to replace as many red indices as possible with green indices, while avoiding bad replacements that harms image quality. To achieve controllable watermark strength that balances between watermark strength and image quality, we propose a confidence-guided index replacement strategy. Specifically, we use the classification probability of an index predicted by the autoregressive model as the confidence measure and calculate relative confidence (will be detailed in Eq. (3)). The greater the relative confidence, the larger the gap between the red index and the paired green index at the current index position. Replacing these indices with significant gaps can lead to a noticeable decline in image quality. Based on the relative confidence distribution of two indices within each pair, we calculate a quantile as the replacement threshold to control watermark strength. For a given replacement threshold, we prioritize replacing index pairs with smaller relative confidence to balance watermark strength and image quality.

When the autoregressive model generates the $k$-th red index $\text{Idx}_k$, we record the classification probability $P(\text{Idx}_k)$ for $\text{Idx}_k$ and the classification probability $P(\text{Idx}'_k)$ for its paired green index $\text{Idx}'_k$. After the autoregressive model generates all indices, we obtain the confidence set for red indices, $\text{conf} = \{P(\text{Idx}_1), P(\text{Idx}_2), \ldots, P(\text{Idx}_{N_{\text{red}}})\}$, and the confidence set for paired green indices, $\text{conf}' = \{P(\text{Idx}'_1), P(\text{Idx}'_2), \ldots, P(\text{Idx}'_{N_{\text{red}}})\}$, where $N_{\text{red}}$ represents the total number of red indices. Based on these two sets of confidence, we calculate the relative confidence for each index pair:

$$\text{relative-conf}_k = \log(P(\text{Idx}_k)/P(\text{Idx}'_k)), \tag{3}$$

where $k$ represents the relative confidence of the $k$-th index pair. We achieve controllable watermark strength by setting a distribution quantile, with the relative confidence distribution illustrated in Figure 3. Specifically, when replacing red indices with paired green indices, we only replace index pairs on the left side of the quantile. Therefore, when the quantile is set to 0%, the model does not perform red index replacement, resulting in a non-watermarked image. When the quantile is set to 100%, the model replaces all red indices with green indices from their index pairs. For other quantile values, the model prioritizes replacing red indices with green indices of lower relative confidence. Additionally, the confidence distribution exhibits a characteristic pattern of low density at both ends and high density in the middle, making index pairs with high relative

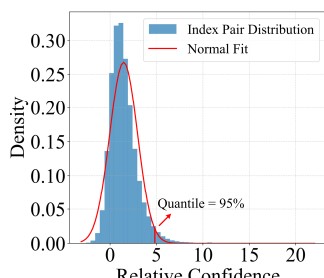

Figure 3: Index pair distribution of one hundred generated images.

confidence more likely to be filtered out. For example, by simply setting the quantile to 95%, we can filter out all index pairs with a relative confidence greater than 5. This design not only achieves controllability of watermark strength but also optimizes the balance between watermark strength and image quality by "filtering" index pairs with high relative confidence.

## 3.2 WATERMARK VERIFICATION

**Statistical Probability-Based Watermark Verification**    Since the red and green indices are randomly assigned, the proportion of green indices in an image without a watermark is approximately 50%, while the proportion in an ideal watermarked image approaches 100%. In fact, the process of autoregressive image generation can be regarded as $N_{\text{Idx}}$ independent Bernoulli trials with equal probability of taking the value 0 (red index) or 1 (green index), where $N_{\text{Idx}}$ is the total number of indices in an image. According to the Central Limit Theorem (Feller, 1991), when $N_{\text{Idx}}$ is sufficiently large, the sample mean follows a normal distribution. Thus, we can calculate the confidence interval CI for the mean of $N_{\text{Idx}}$ trials at a confidence level of $1 - \beta$ as follows:

$$\text{CI} = \left(0.5 - \frac{z_{\beta/2}}{2\sqrt{N_{\text{Idx}}}}, \ 0.5 + \frac{z_{\beta/2}}{2\sqrt{N_{\text{Idx}}}}\right), \tag{4}$$

where $z_{\beta/2}$ represents the two-tailed critical value of the standard normal distribution. After calculating the confidence interval, we can use the right endpoint of the confidence interval as a decision threshold. If the proportion of green indices in an image is below the threshold, the image is classified as a non-watermarked image; otherwise, it is classified as a watermarked image.

**Index Encoder**    VQ-VAE is designed solely for pixel-level reconstruction. However, the objective of the watermark validator in this paper is index reconstruction. In practice, the autoregressively

Figure 4: Training of Index Encoder. The Encoder, Codebook, and Decoder are frozen while the Index Encoder is updated to achieve accurate index reconstruction.

generated indices $\mathbf{Idx} = \{\text{Idx}_1, \text{Idx}_2, \ldots, \text{Idx}_{N_{\text{Idx}}}\}$ are processed through the decoder to produce the image $I$. However, the reconstructed indices $\mathbf{Idx}^{rc}$ obtained by feeding $I$ into the encoder and vector quantization module, may differ from the original $\mathbf{Idx}$. As illustrated in Figure 6(b), the original VQ-VAE encoder struggles to accurately identify watermarks in high-confidence scenarios. Therefore, we propose an *Index Encoder* to assist users in high-confidence scenarios in achieving high-precision index reconstruction.

As shown in Figure 4, we freeze the encoder, codebook, and decoder, and retrain a new encoder, termed *Index Encoder*, with the goal of achieving accurate reconstruction of the indices $\mathbf{Idx}$. We first input the original image $I^{real}$ into the encoder and decoder to obtain the vector $z$ and image $I$. Then, we input $I$ into the Index Encoder and decoder to obtain the reconstructed vector $z^{rc}$ and reconstructed image $I^{rc}$. The optimization of the Index Encoder is performed by minimizing two loss terms: (1) the mean squared error (MSE) between the vector $z$ and the vector $z^{rc}$, and (2) the MSE between the image $I$ and the reconstructed image $I^{rc}$:

$$\mathcal{L}_{encoder} = \|z^{rc} - z\|_2^2 + \gamma \|I^{rc} - I\|_2^2, \tag{5}$$

where $\gamma$ represents the weight hyperparameter.

**Cropped Image Watermark Verification**    Although the red–green index watermark itself is highly robust, as demonstrated in Table 1, the VQ-VAE encoding paradigm is inherently vulnerable to cropping attacks. During index reconstruction, VQ-VAE first divides an image into fixed-size, non-overlapping patches (*e.g.*, $8 \times 8$ pixels) and independently encodes each patch to retrieve the index. Then, even a slight crop to the image can drastically alter the patch composition. For instance, consider cropping an image such that the new top-left corner lands at position (4, 3) within the neighborhood of the original patch. Such a tiny shift entails that every subsequent pixel now belongs to completely different spatial segments compared to the original image. When encoded, these reconfigured patches will lead to entirely different codebook indices, thereby significantly weakening watermark-verification robustness.

To address this weakness, we propose traversing every pixel in the local image block of the cropped image to achieve alignment of the local image blocks. Taking an $8 \times 8$ pixel block as an example, suppose the top-left corner of the cropped image is originally located at (4, 3) of a block. We enumerate the cropped image to traverse all pixel positions within the first local image block. That is, the top-left corner of the cropped image moves from (1, 1) to (8, 8), stopping after enumerating 64 candidate images. For each candidate, we calculate its green index rate. As long as the green index rate of one of the candidates reaches the decision threshold, the image is considered to contain a watermark. For example, as the image moves from (1, 1), the green index rate remains close to 50%. However, when the top-left corner of the cropped image moves to (4, 3), the green index rate reaches 100%, indicating the presence of a watermark. The example can be found in the Appendix C.3.

## 4 EXPERIMENTS

**Model and Benchmark**    We conduct experiments using a SOTA autoregressive image generation model, LlamaGen (Sun et al., 2024a). We employ ACC to measure watermark verification performance, utilizing PSNR, SSIM and MSSIM (Wang et al., 2004) to quantify pixel-level differences between watermarked and original images. Furthermore, we assess the fidelity of watermarked image distributions using FID (Heusel et al., 2017), and evaluate text-image alignment with CLIP scores. We compare IndexMark with five baseline approaches: three post-processing methods including

Table 1: Comparison of IndexMark with LLM watermarking methods in terms of quality.

| Model | Method | PSNR ↑ | SSIM ↑ | MSSIM ↑ | CLIP ↑ | FID ↓ |
|---|---|---|---|---|---|---|
| **MSCOCO Dataset** | | | | | | |
| LlamaGen (AR) (256 × 256) | W/o watermark | ∞ | 1.000 | 1.000 | 0.328 | 25.53 |
| | KGW | 10.16 | 0.304 | 0.132 | **0.326** | 25.81 |
| | STA-1 | 10.17 | 0.309 | 0.123 | **0.326** | 25.67 |
| | IndexMark | **23.54** | **0.838** | **0.930** | 0.326 | **24.73** |
| LlamaGen (AR) (512 × 512) | W/o watermark | ∞ | 1.000 | 1.000 | 0.282 | 54.57 |
| | KGW | 9.43 | 0.248 | 0.113 | 0.280 | 54.87 |
| | STA-1 | 10.01 | 0.305 | 0.133 | **0.281** | 54.64 |
| | IndexMark | **24.15** | **0.838** | **0.930** | 0.281 | **54.35** |
| **ImageNet Dataset** | | | | | | |
| LlamaGen (AR) (256 × 256) | W/o watermark | ∞ | 1.000 | 1.000 | 0.289 | 15.08 |
| | KGW | 9.78 | 0.262 | 0.131 | **0.288** | 15.15 |
| | STA-1 | 9.73 | 0.261 | 0.157 | **0.288** | 15.12 |
| | IndexMark | **23.86** | **0.738** | **0.903** | 0.288 | **13.89** |
| LlamaGen (AR) (384 × 384) | W/o watermark | ∞ | 1.000 | 1.000 | 0.287 | 12.65 |
| | KGW | 9.87 | 0.291 | 0.162 | **0.286** | 12.78 |
| | STA-1 | 9.91 | 0.312 | 0.174 | **0.286** | 12.67 |
| | IndexMark | **25.45** | **0.783** | **0.913** | 0.286 | **11.81** |

Table 2: Comparison of IndexMark with post-processing and autoregressive watermarking methods in terms of robustness against various attacks.

| Method | | Clean | Blur | Noise | JPEG | Bright | Erase | Crop | Rot | Avg |
|---|---|---|---|---|---|---|---|---|---|---|
| **MSCOCO Dataset** | | | | | | | | | | |
| Post-processing | DwtDct | 0.603 | 0.501 | 0.607 | 0.500 | 0.571 | 0.567 | 0.500 | 0.500 | 0.542 |
| | DwtDctSvd | 0.996 | 0.982 | 0.994 | 0.963 | 0.556 | 0.994 | 0.500 | 0.502 | 0.810 |
| | RivaGAN | 0.930 | 0.919 | 0.929 | 0.727 | 0.862 | 0.847 | 0.500 | 0.519 | 0.778 |
| Auto-regressive | KGW | **1.000** | 0.968 | 0.989 | 0.968 | 0.971 | 0.992 | 0.500 | 0.943 | 0.915 |
| | STA-1 | **1.000** | 0.965 | 0.991 | 0.971 | 0.975 | 0.989 | 0.500 | 0.952 | 0.917 |
| | IndexMark | **1.000** | **0.991** | **0.995** | **0.978** | **0.988** | **0.997** | **0.998** | **0.973** | **0.989** |
| **ImageNet Dataset** | | | | | | | | | | |
| Post-processing | DwtDct | 0.583 | 0.501 | 0.588 | 0.500 | 0.584 | 0.568 | 0.500 | 0.500 | 0.540 |
| | DwtDctSvd | 0.994 | 0.991 | 0.989 | 0.960 | 0.552 | 0.994 | 0.500 | 0.502 | 0.809 |
| | RivaGAN | 0.951 | 0.930 | 0.950 | 0.746 | 0.919 | 0.914 | 0.500 | 0.518 | 0.803 |
| Auto-regressive | KGW | **1.000** | 0.999 | 0.998 | **1.000** | 0.992 | 0.999 | 0.500 | 0.988 | 0.933 |
| | STA-1 | **1.000** | 0.997 | 0.999 | **1.000** | 0.994 | 0.998 | 0.500 | 0.987 | 0.933 |
| | IndexMark | **1.000** | **1.000** | **1.000** | **1.000** | **0.998** | **1.000** | **0.998** | **0.989** | **0.997** |

DwtDct (Al-Haj, 2007), DwtDctSvd (Navas et al., 2008), and RivaGAN (Zhang et al., 2019), and two LLM methods including KGW (Kirchenbauer et al., 2023) and STA-1 (Mao et al., 2025).[2]

**Implementation Details** For the construction of index pairs, we set top-K pruning with $K = 10$. For the text-conditioned generation task, we set top-K sampling with $K = 1000$, CFG-scale to 7.5, and downsample-size to 16. For the class-conditioned generation task, we set top-K sampling with $K = 2000$, CFG-scale to 4.0, downsample-size to 16, and default to a full-green index watermark. For the Index Encoder, we conducted training using the MS-COCO-2017 training dataset (Lin et al., 2014) and ImageNet-1k validation dataset (Deng et al., 2009), employing the Adam optimizer (Kingma & Ba, 2014) with a learning rate of 1e-5, and set $\gamma$ to 0.5. All experiments run on NVIDIA A100 GPUs.

## 4.1 IMAGE QUALITY AND WATERMARK ROBUSTNESS

**Image Quality** As shown in Table 1, our method demonstrates significant advantages in image quality. Direct application of LLM watermarking methods causes severe quality degradation, with PSNR dropping sharply to approximately 10 dB while FID scores deteriorate significantly. This

---

[2]More details about how the evaluation metrics are computed can be found in the Appendix D.

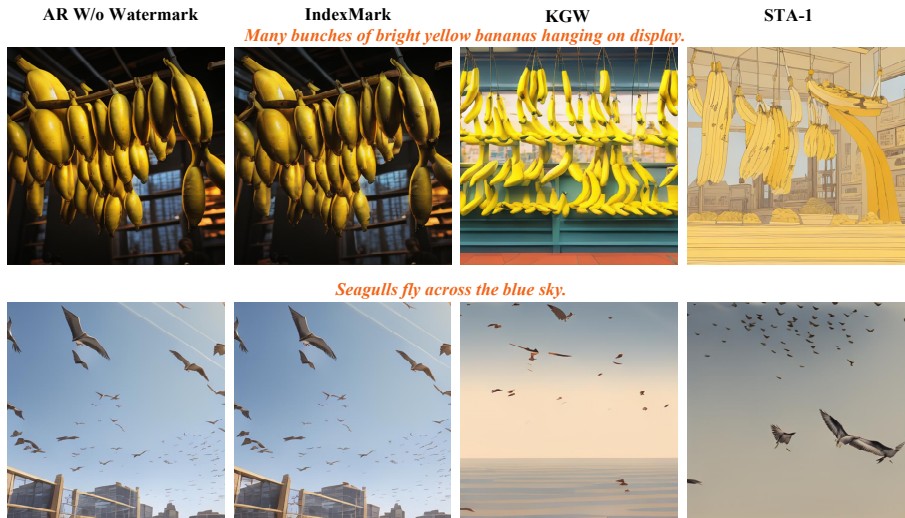

**AR W/o Watermark**  **IndexMark**  **KGW**  **STA-1**

*Many bunches of bright yellow bananas hanging on display.*

*Seagulls fly across the blue sky.*

Figure 5: IndexMark vs. LLM Watermarkings. LLM watermarkings embed watermarks by interfering with the image generation path, and their inherent bias list may lead to degraded image quality. In contrast, IndexMark employs a *match-then-replace* strategy for watermark embedding, ensuring both image quality and output diversity.

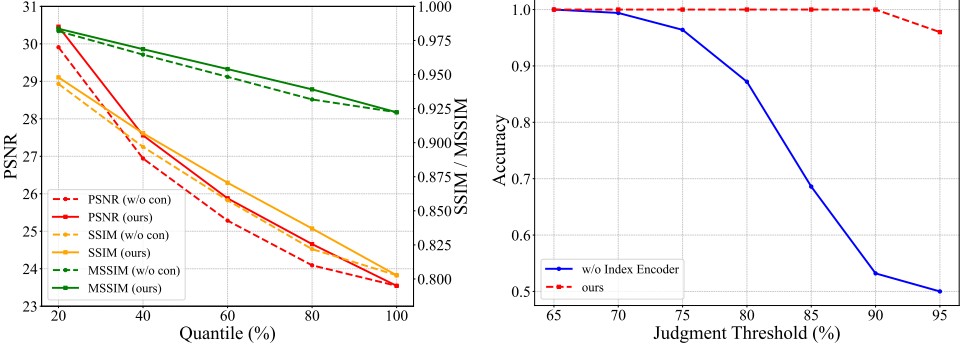

(a) Evaluation on confidence-guided replacement.   (b) Evaluation on Index Encoder.

Figure 6: Ablation results on confidence-guided index replacement and Index Encoder.

confirms that sampling strategies designed for large text vocabularies, when applied to compact image codebooks, disrupt the image generation process—not only impairing content diversity but also introducing visual artifacts like unnatural color patches. Figure 5 shows the negative impact of LLM watermarking on image quality, greatly compromising watermark imperceptibility. In contrast, IndexMark excels in preserving image quality. This advantage stems from its Construction of Index Pairs technique, resolving perceptibility issues via optimal matching, while its Confidence-Guided Index Replacement ensures image quality. Additionally, IndexMark prevents disrupting the generation process, ensuring diverse image outputs. More results can be found in the Appendix E.4.

**Robustness**   To evaluate the robustness of IndexMark, we select seven common data augmentations as attack methods. These include Gaussian blur with a kernel size of 11, Gaussian noise with a standard deviation of $\sigma = 0.01$, JPEG with a quality factor of 70, color jitter with brightness set to 0.5, random erasing of 10% of the region, random cropping of 75%, and random rotation. We select the right endpoint of the confidence interval at a 99.9% confidence level as the threshold for watermark determination. Table 2 presents the results at $256 \times 256$ resolution, demonstrating IndexMark's exceptional robustness. More results and details of the rotation verification are in Appendix E.3.

### 4.2   ABLATION STUDY AND FURTHER ANALYSES

**Confidence-Guided Index Replacement**   We substitute the confidence-guided method with random index selection based on watermark strength. The results shown in Figure 6(a) justify the effectiveness of our design as the PSNR scores of random index selection are significantly lower than IndexMark.

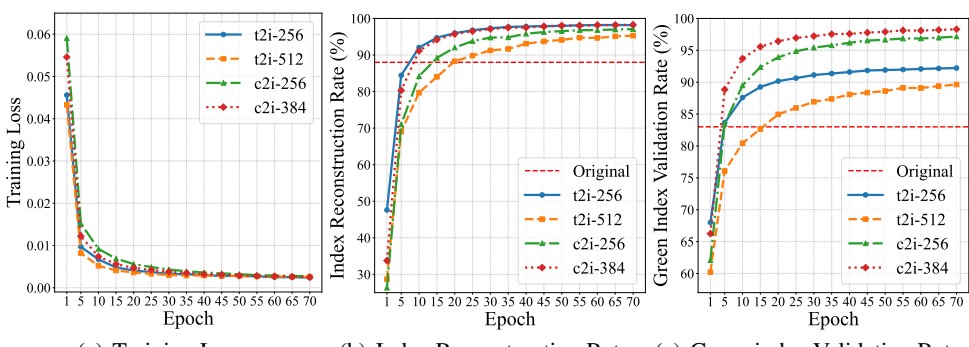

(a) Training Loss.        (b) Index Reconstruction Rate.    (c) Green index Validation Rate.

Figure 7: Images showing the variation of training loss, index reconstruction rate, and green index verification rate of the Index Encoder with respect to epochs.

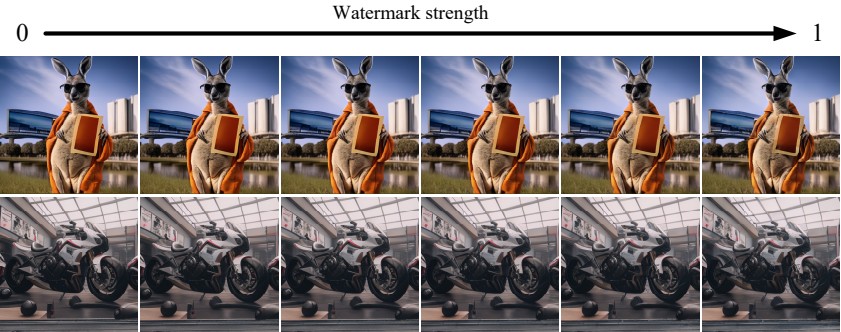

Figure 8: Generated images under different watermark strengths.

**Index Encoder**    We compare watermark verification rates with and without Index Encoder at different confidence levels on $256 \times 256$ resolution images. As shown in Figure 6(b), differences are minimal at lower confidence levels, but at higher levels, Index Encoder significantly improves verification rates. The results without using the Index Encoder can be found in the Appendix E.2.

**Index Reconstruction**    To investigate whether the Index Encoder improves index reconstruction capability, we conduct index-to-index reconstruction experiments at a resolution of 256 across multiple epochs using the Index Encoder, as well as validation experiments on pure green index images. Additionally, Figure 7(a) illustrates the training loss across multiple resolutions. As shown in Figure 7(b), after only 20 epochs of training, the index reconstruction capability of the Index Encoder surpasses that of the original encoder. Furthermore, as depicted in Figure 7(c), the Index Encoder's validation capability for images with all indices being green significantly exceeds that of the original encoder, allowing users to verify watermarks with a higher confidence level.

**Watermark Strength**    We explore the impact of watermark strength on images. The qualitative results, as shown in Figure 8, indicate that an increase in IndexMark watermark strength does not cause noticeable changes in image quality.

## 5   CONCLUSION

This paper proposes IndexMark, the first *training-free* watermarking method for autoregressive image generation models. IndexMark carefully selects watermark tokens from the codebook based on token similarity and promotes the use of watermark tokens through token replacement, thereby embedding the watermark in the image. We believe that our method offers a novel perspective for watermark design in autoregressive image generation models.

# 6 ETHICS STATEMENT

We declare that this research fully adheres to ethical guidelines. The study focuses on watermarking for autoregressive image models and does not involve any human subjects. All datasets used are publicly available and contain no privacy-sensitive content.

# 7 REPRODUCIBILITY STATEMENT

Our code will be made publicly available at a later stage to ensure the reproducibility of the experiments. All datasets involved are publicly accessible.

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

APPENDIX

## A  THE USE OF LLM

We strictly limit the use of large language models to grammar checking and selective word translation. We avoid any intensive usage that could potentially lead to academic misconduct.

## B  FUTURE WORK AND SOCIAL IMPACT

### B.1  FUTURE WORK

The verification of IndexMark watermark relies on the index reconstruction capability of the VQ-VAE model. A more robust encoder can enhance the robustness of our method, such as index reconstruction based on image semantics (Yu et al., 2024). Additionally, our current match-then-replace method uses simple pairwise matching. By exploring diverse matching methods, we can further leverage the redundancy of the codebook, thereby improving the quality of the watermarked images.

### B.2  SOCIAL IMPACT

With the rapid advancement of autoregressive image generation models, developers have the responsibility and obligation to ensure the safety of these models. We provide developers with an efficient and effective method to help them counteract the misuse of models, marking a step towards responsible AI in autoregressive image generation models.

## C  MODEL DETAILS

### C.1  VQ-VAE AND AUTOREGRESSIVE IMAGE GENERATION

**VQ-VAE**  The Vector Quantized Variational Autoencoder (VQ-VAE) provides a framework for encoding images into a discrete latent representation. Given an input image $x \in \mathbb{R}^{H \times W \times 3}$, the encoder produces a continuous latent feature map:

$$z = \text{encoder}(x) \in \mathbb{R}^{h \times w \times d}. \tag{6}$$

For every spatial location $(i, j)$ we find the nearest entry in the VQ-VAE's codebook $\mathcal{C} = \{e_1, e_2, \ldots, e_K\} \subset \mathbb{R}^d$:

$$k_{ij} = \underset{k \in \{1, \ldots, K\}}{\arg\min} \|z_{ij} - e_k\|_2, \qquad z_{ij}^q = e_{k_{ij}}, \tag{7}$$

where $k_{ij}$ is a discrete index, and $z_{ij}^q$ is the corresponding quantised vector. By flattening the quantized vector $z^q$, a sequence of discrete tokens $T = \{T_1, T_2, \ldots, T_{h \times w}\}$ is obtained, where each token $T_i$ represents an index in the codebook $\mathcal{C}$. During the reconstruction phase, the quantized latent vector $z^q$ is retrieved using the token indices and the codebook. This vector is then passed through a decoder to reconstruct the original image: $\hat{x} = \text{decoder}(z^q)$. During the training phase, the model is constrained by the image reconstruction loss, codebook loss, and commitment loss, defined as:

$$\mathcal{L} = \|x - \hat{x}\|_2^2 + \beta\|q - \text{sg}[z]\|_2^2 + \gamma\|z - \text{sg}[q]\|_2^2, \tag{8}$$

where sg denotes the stop gradient operation.

**Autoregressive Image Generation**  The autoregressive model defines the generation process as the prediction of the next token:

$$p(\mathbf{x}) = \prod_{i=1}^{n} p(x_i \mid x_1, x_2, \ldots, x_{i-1}) = \prod_{i=1}^{n} p(x_i \mid x_{<i}). \tag{9}$$

In autoregressive image generation, $x_i$ denotes the image token in the discrete latent space, and the image generation process can be formulated as:

$$p(\mathbf{q}) = \prod_{i=1}^{h \times w} p(q_i \mid q_{<i}, c), \tag{10}$$

where $q_i$ denotes the discretized image token, $c$ denotes the embedding of the class label or the text, and $h \times w$ represents the total number of image tokens. During the training phase, the model is trained by maximizing the likelihood of the observed token sequences:

$$L_{\text{train}} = -\log p(\mathbf{q}) = -\sum_{i=1}^{h \times w} \log p\big(q_i \mid q_{<i}, c\big). \tag{11}$$

During inference, the model generates the sequence of token indices autoregressively by sampling each next index. Once the full sequence of image token indices is produced, the codebook is used to reconstruct the latent vector $z_q$ from those indices, and $z_q$ is then fed into the VQ-VAE decoder to synthesize the final image.

## C.2 BLOSSOM

The core principle of the Blossom algorithm is to iteratively approach the optimal matching by dynamically handling odd-length cycle structures within the graph. Its key steps are as follows:

- **Blossom Shrinking**: When the algorithm verifies an odd cycle, it contracts the cycle into a super vertex, preserving the connections between the cycle and external vertices, thereby simplifying the complex structure into a recursively manageable subgraph.

- **Augmenting Path Search**: The current matching is expanded by traversing a path that alternates between matched and unmatched edges. During each expansion, the matching status of the edges on the path is flipped to increase the total weight.

- **Dual Variable Adjustment**: Utilizing the duality theory of linear programming, the potentials of vertices and odd sets are adjusted to ensure that each operation converges toward maximizing the total weight.

The pseudocode of the Blossom Algorithm is shown in Algorithm 1.

---

**Algorithm 1:** Blossom Algorithm

---

**Input:** Graph $G = (V, E)$, edge weights $w : E \to \mathbb{R}$
**Output:** Maximum-weight perfect matching $M \subseteq E$
```
// Initialize
```
1 $M \leftarrow \varnothing$
2 $y(v) \leftarrow \frac{1}{2} \max_{e \in \delta(v)} w(e)$ for all $v \in V$
3 $\mathcal{B} \leftarrow \varnothing$
4 **while** *M is not perfect* **do**
5     Search for augmenting paths via BFS/DFS        // Build alternating trees
6     **if** *any odd-length cycle B found* **then**
```
            // Blossom Shrinking
```
7        Contract $B$ into super-node $b$
8        Update $\mathcal{B} \leftarrow \mathcal{B} \cup \{b\}$
9        Adjust dual variables $y$ and $z_B$ for $b$        // Maintain LP feasibility
10     **if** *augmenting path P found* **then**
```
            // Augment matching
```
11        $M \leftarrow M \oplus P$        // Symmetric difference
12
13        Expand blossoms in $\mathcal{B}$ along $P$        // Restore original graph
14
15        Reset search structures
```
        // Dual Variable Adjustment
```
16     Compute $\delta = \min\{\text{slack}(e) \mid e \in E\}$
17     Update $y(v) \leftarrow y(v) \pm \delta$ and $z_B \leftarrow z_B + 2\delta$        // Converge to optimality
18 **return** $M$

---

## C.3 WATERMARK VERIFICATION ON CROPPED IMAGE

In the Figure 9, we show watermark verification process on a cropped image. As an example, with an $8 \times 8$ input image and using a patch side length of 2, by traversing the first image patch, the watermark in the cropped image can be successfully verified with at most $2 \times 2$ checks.

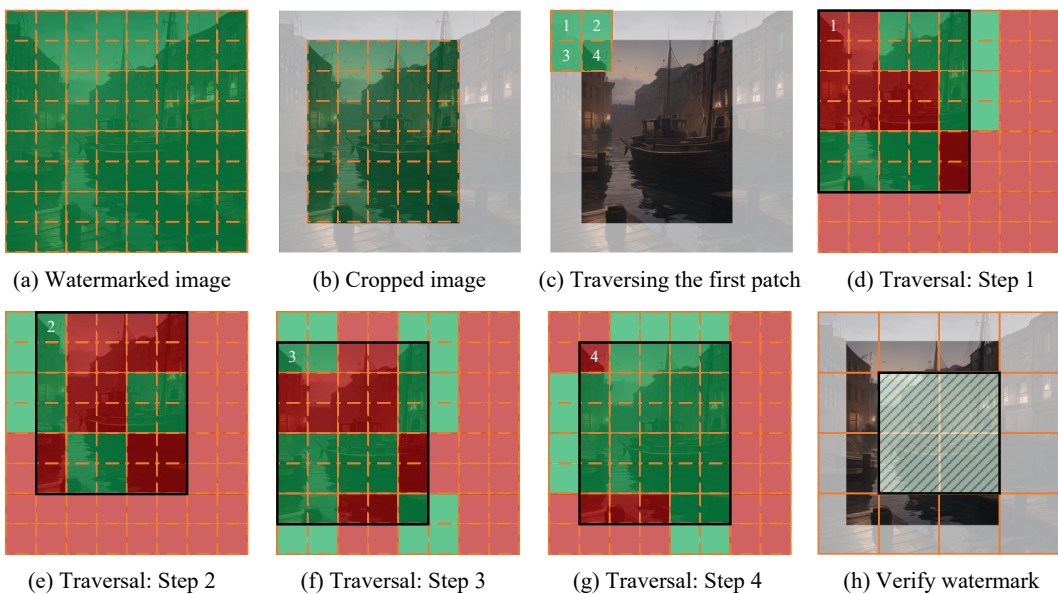

| (a) Watermarked image | (b) Cropped image | (c) Traversing the first patch | (d) Traversal: Step 1 |

| (e) Traversal: Step 2 | (f) Traversal: Step 3 | (g) Traversal: Step 4 | (h) Verify watermark |

Figure 9: Visualization of the traversal process for watermark verification on the cropped image.

## C.4 CAPACITY ESTIMATION

Given a codebook of size $N$, we can construct $M = N/2$ index pairs, which means that each user's key is a binary string of length $M$. For any two users $A$ and $B$, we denote $p_{A|B}$ as the green index rate when using key $B$ to verify an image generated with key $A$, and let $d$ represent the Hamming distance between the two keys. Let $n$ denote the number of indices in the image. $p_{A|B}$ is a random variable, and according to the Central Limit Theorem, $p_{A|B}$ approximately follows a normal distribution $N(\mu, \sigma^2)$. We obtain that the expectation of $p_{A|B}$ is $(M - d)/M$, and its variance is $(1 - d/M)(d/M)/n$.

To prevent watermark confusion between user $A$ and $B$, it is necessary to ensure that the measured value of $p_{A|B}$ is leass than the watermark decision threshold. Therefore, we need to ensure that the vast majority of samples from the distribution of $p_{A|B}$ are below the watermark threshold $x$ which can be approximated by $\mu + 3\sigma < x$. According to this inequality, the minimum Hamming distance $d_{min}$ can be determined. Next, we calculate the maximum number of keys $K$ in the binary space under the constraint of the minimum Hamming distance. According to the Hamming bound, we have:

$$K = \left\lfloor \frac{2^M}{\sum_{i=0}^{t} \binom{M}{i}} \right\rfloor, \quad \text{where} \quad t = \left\lfloor \frac{d_{\min} - 1}{2} \right\rfloor. \tag{12}$$

When the watermark threshold is 0.615 (the false positive rate is at the $10^{-4}$ level) and the image size is 256, the maximum number of user keys is approximately $2^{1696}$.

## D EXPERIMENTAL DETAILS

### D.1 DETAILS ABOUT EVALUATION METRICS

**FID** For text-to-image tasks, we generate 5,000 images to evaluate the Fréchet Inception Distance (FID) score (Heusel et al., 2017) on the MS-COCO-2017 training dataset. For class-conditioned

image generation tasks, we generate 10,000 images to evaluate the FID score on the ImageNet-1k validation dataset.

**CLIP Score**   We use OpenCLIP-ViT model (Cherti et al., 2023) to compute the CLIP score (Radford et al., 2021) between generated images and their corresponding text prompts. For class-conditioned generation, we use "a photo of category" as the input.

### D.2   DETAILS OF THE THRESHOLD FOR WATERMARK DETERMINATION

For $256 \times 256$ and $384 \times 384$ resolutions, we select a green index rate of 0.615 near the 99.9% confidence level as the determination threshold. For $512 \times 512$ resolution, we choose a green index rate of 0.60 near the 99.99% confidence level as the determination threshold. Regarding cropping attacks, since the image is reduced to approximately 50% of its original size, we use a higher confidence level to detect the watermark. Specifically, for $512 \times 512$ resolution, we use a green index rate of 0.65 as the determination threshold, while for $256 \times 256$ and $384 \times 384$ resolutions, we adopt 0.7 as the determination threshold.

## E   MORE EXPERIMENTAL RESULTS

### E.1   GREEN INDEX GENERATION

We explored the possibility of generating images using only the green indices from the codebook, referring to this variant as GreenGen. As shown in Figure 10, the watermarked images generated by GreenGen exhibit significant differences compared to the watermark-free images. The quantitative results are shown in Table 3. GreenGen differs significantly from the watermarked image at the pixel level. Although GreenGen achieves a CLIP score similar to that of IndexMark, its performance in terms of FID is not as good as IndexMark. This result indicates that there is a substantial amount of redundancy in the codebook, and our method effectively leverages this redundancy to achieve better watermark embedding while maintaining image quality and content integrity.

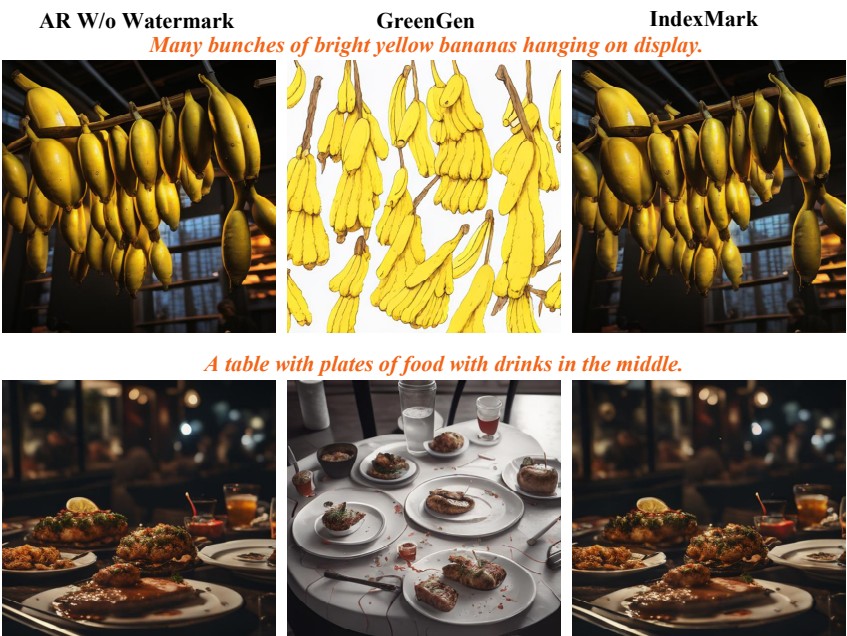

Figure 10: GreenGen vs. IndexMark. GreenGen generates autoregressive images by removing red indices from the codebook and using only green indices, resulting in significant differences between the watermarked images and non-watermarked ones. In contrast, IndexMark achieves smaller differences through a match-then-replace method.

Table 3: Comparison results of image quality between IndexMark and GreenGen.

| Model | Method | PSNR ↑ | SSIM ↑ | MSSIM ↑ | CLIP ↑ | FID ↓ |
|---|---|---|---|---|---|---|
| **MSCOCO Dataset** | | | | | | |
| LlamaGen (AR) (256 × 256) | W/o watermark | ∞ | 1.000 | 1.000 | 0.328 | 26.55 |
| | GreenGen | 9.76 | 0.267 | 0.111 | **0.326** | 26.35 |
| | IndexMark | **23.54** | **0.838** | **0.930** | 0.326 | **24.73** |
| LlamaGen (AR) (512 × 512) | W/o watermark | ∞ | 1.000 | 1.000 | 0.282 | 54.57 |
| | GreenGen | 10.11 | 0.280 | 0.129 | **0.281** | 54.51 |
| | IndexMark | **24.15** | **0.838** | **0.930** | 0.281 | **54.35** |
| **ImageNet Dataset** | | | | | | |
| LlamaGen (AR) (256 × 256) | W/o watermark | ∞ | 1.000 | 1.000 | 0.289 | 15.08 |
| | GreenGen | 9.46 | 0.186 | 0.106 | **0.288** | 15.30 |
| | IndexMark | **23.86** | **0.738** | **0.903** | 0.288 | **13.89** |
| LlamaGen (AR) (384 × 384) | W/o watermark | ∞ | 1.000 | 1.000 | 0.287 | 12.65 |
| | GreenGen | 9.454 | 0.230 | 0.131 | **0.286** | 12.46 |
| | IndexMark | **25.45** | **0.783** | **0.913** | 0.286 | **11.81** |

## E.2 ROBUSTNESS EXPERIMENT WITHOUT THE INDEX ENCODER

Under lower watermark-verification confidence thresholds, we removed the Index Encoder (w/o IE) and conducted robustness experiments. As shown in Table 4, even at low confidence settings, the model without the Index Encoder maintains strong robustness, thereby reducing training costs for users with less stringent security requirements.

Table 4: Comparison of ACC across different watermarking methods under various attacks. Clean indicates watermark verification results on unaltered images, while Avg represents the average accuracy across all attack scenarios.

| Model | Method | Clean | Blur | Noise | JPEG | Bright | Erase | Crop | Rot | Avg |
|---|---|---|---|---|---|---|---|---|---|---|
| **MSCOCO Dataset** | | | | | | | | | | |
| LlamaGen (AR) (256 × 256) | w/o IE | **1.000** | 0.972 | 0.990 | 0.970 | 0.974 | **0.997** | 0.917 | 0.949 | 0.970 |
| | IndexMark | **1.000** | **0.991** | **0.995** | **0.978** | **0.988** | **0.997** | **0.998** | **0.973** | **0.989** |
| LlamaGen (AR) (512 × 512) | w/o IE | **1.000** | 0.969 | 0.992 | 0.980 | 0.981 | **0.992** | 0.939 | 0.992 | 0.979 |
| | IndexMark | **1.000** | **0.988** | **0.994** | **0.984** | **0.989** | **0.992** | **0.993** | **0.995** | **0.991** |
| **ImageNet Dataset** | | | | | | | | | | |
| LlamaGen (AR) (256 × 256) | w/o IE | **1.000** | **1.000** | **1.000** | **1.000** | 0.995 | **1.000** | 0.996 | 0.988 | 0.996 |
| | IndexMark | **1.000** | **1.000** | **1.000** | **1.000** | **0.998** | **1.000** | **0.998** | **0.989** | **0.997** |
| LlamaGen (AR) (384 × 384) | w/o IE | **1.000** | 0.999 | 0.999 | **1.000** | 0.994 | 0.999 | 0.905 | 0.975 | 0.983 |
| | IndexMark | **1.000** | **1.000** | **1.000** | **1.000** | **0.998** | **1.000** | **0.993** | **0.980** | **0.995** |

## E.3 ROBUSTNESS TESTING

**Watermark Verification Details Against Rotation Attacks**    We rotated the watermarked images by 0, 0.1, 0.2, 0.3, 0.4, and 0.5 degrees respectively and calculated the green index ratio for each. The results are shown in Table 5. We found that the watermark could still be verified even after a 0.5-degree rotation. Therefore, we iteratively rotate the image in 1-degree increments until the green index ratio meets the watermark threshold, at which point the rotation stops. If no watermark is verified after a full rotation (360 degrees), it indicates that the image contains no watermark. Moreover, to accelerate the verification process for both 512 × 512 and 384 × 384 resolutions, we compute the green index ratio solely for the central 256 indices.

Table 5: Watermark Verification Sensitivity to Rotation

| Rotation Angle (°) | 0.0 | 0.1 | 0.2 | 0.3 | 0.4 | 0.5 |
|---|---|---|---|---|---|---|
| Green Index Ratio (%) | 100.0 | 84.7 | 81.5 | 78.1 | 77.2 | 75.7 |

Table 6: Comparison of IndexMark with post-processing and autoregressive watermarking methods in terms of robustness against various attacks.

| Resolution | Method | Clean | Blur | Noise | JPEG | Bright | Erase | Crop | Rot | Avg |
|---|---|---|---|---|---|---|---|---|---|---|
| | | | | **MSCOCO Dataset** | | | | | | |
| $256 \times 256$ | DwtDct | 0.603 | 0.501 | 0.607 | 0.500 | 0.571 | 0.567 | 0.500 | 0.500 | 0.542 |
| | DwtDctSvd | 0.996 | 0.982 | 0.994 | 0.963 | 0.556 | 0.994 | 0.500 | 0.502 | 0.810 |
| | RivaGAN | 0.930 | 0.919 | 0.929 | 0.727 | 0.862 | 0.847 | 0.500 | 0.519 | 0.778 |
| | KGW | **1.000** | 0.968 | 0.989 | 0.968 | 0.971 | 0.992 | 0.500 | 0.943 | 0.915 |
| | STA-1 | **1.000** | 0.965 | 0.991 | 0.971 | 0.975 | 0.989 | 0.500 | 0.952 | 0.917 |
| | IndexMark | **1.000** | **0.991** | **0.995** | **0.978** | **0.988** | **0.997** | **0.998** | **0.973** | **0.989** |
| $512 \times 512$ | DwtDct | 0.741 | 0.512 | 0.739 | 0.500 | 0.680 | 0.734 | 0.500 | 0.500 | 0.612 |
| | DwtDctSvd | 0.999 | 0.990 | **0.998** | **0.988** | 0.673 | **0.998** | 0.500 | 0.501 | 0.830 |
| | RivaGAN | 0.973 | 0.967 | 0.970 | 0.900 | 0.930 | 0.945 | 0.958 | 0.529 | 0.896 |
| | KGW | **1.000** | 0.966 | 0.991 | 0.982 | 0.982 | 0.986 | 0.928 | 0.993 | 0.978 |
| | STA-1 | **1.000** | 0.968 | 0.989 | 0.978 | 0.980 | 0.987 | 0.914 | 0.989 | 0.975 |
| | IndexMark | **1.000** | **0.988** | 0.994 | 0.984 | **0.989** | 0.992 | **0.993** | **0.995** | **0.991** |
| | | | | **ImageNet Dataset** | | | | | | |
| $256 \times 256$ | DwtDct | 0.583 | 0.501 | 0.588 | 0.500 | 0.584 | 0.568 | 0.500 | 0.500 | 0.540 |
| | DwtDctSvd | 0.994 | 0.991 | 0.989 | 0.960 | 0.552 | 0.994 | 0.500 | 0.502 | 0.809 |
| | RivaGAN | 0.951 | 0.930 | 0.950 | 0.746 | 0.919 | 0.914 | 0.500 | 0.518 | 0.803 |
| | KGW | **1.000** | 0.999 | 0.998 | **1.000** | 0.992 | 0.999 | 0.500 | 0.988 | 0.933 |
| | STA-1 | **1.000** | 0.997 | 0.999 | **1.000** | 0.994 | 0.998 | 0.500 | 0.987 | 0.933 |
| | IndexMark | **1.000** | **1.000** | **1.000** | **1.000** | **0.998** | **1.000** | **0.998** | **0.989** | **0.997** |
| $384 \times 384$ | DwtDct | 0.720 | 0.521 | 0.725 | 0.500 | 0.780 | 0.696 | 0.500 | 0.500 | 0.617 |
| | DwtDctSvd | 0.999 | 0.990 | 0.999 | 0.542 | 0.664 | 0.999 | 0.500 | 0.500 | 0.773 |
| | RivaGAN | 0.966 | 0.947 | 0.964 | 0.846 | 0.949 | 0.999 | 0.953 | 0.527 | 0.893 |
| | KGW | **1.000** | 0.998 | 0.999 | **1.000** | 0.992 | **1.000** | 0.913 | 0.976 | 0.984 |
| | STA-1 | **1.000** | 0.997 | 0.999 | **1.000** | 0.993 | 0.999 | 0.901 | 0.973 | 0.982 |
| | IndexMark | **1.000** | **1.000** | **1.000** | **1.000** | **0.998** | **1.000** | **0.993** | **0.980** | **0.995** |

**Robustness Results at Multiple Resolutions**  Robustness test results across different resolutions are shown in Table 6, demonstrating our method's strong robustness.

### E.4  MORE QUALITATIVE RESULTS

**W/o Watermark**  **IndexMark**

*Plate of food with gravy on mesh table with knife.*

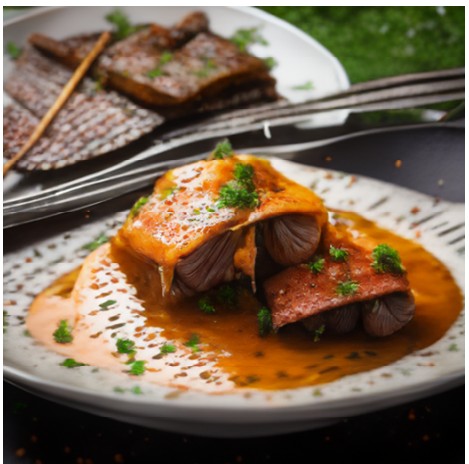 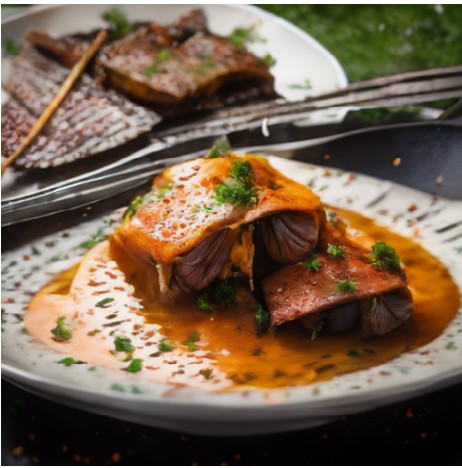

*A black and white chicken is walking through tall plants.*

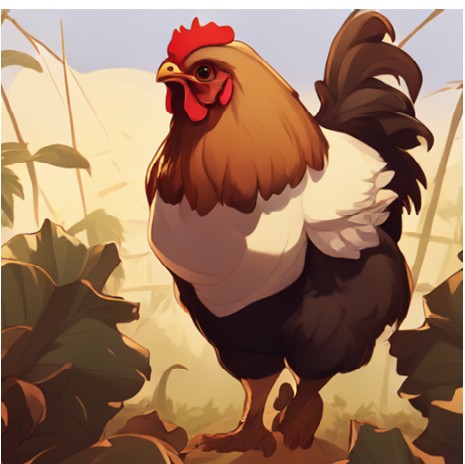 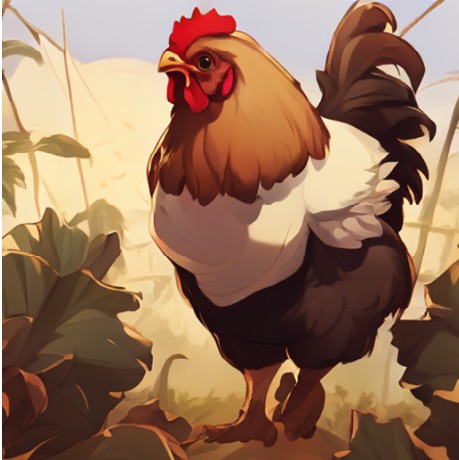

*A cat curled up in a box with a Pirates Hat on.*

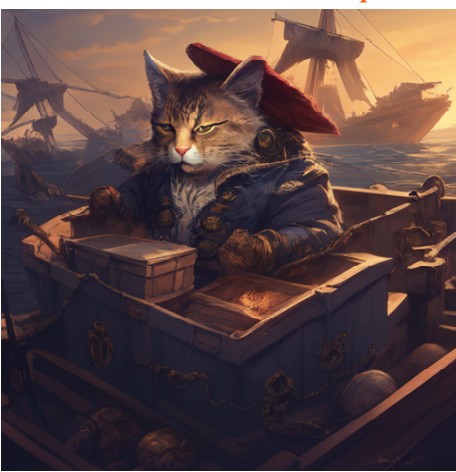 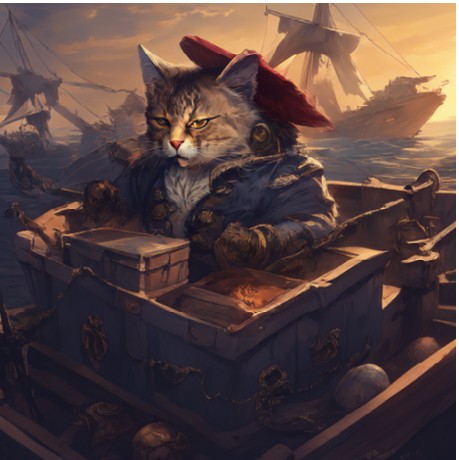

Figure 11: More qualitative comparison results between non-watermarked images and IndexMark watermarked images.

**W/o Watermark**  **IndexMark**

*A gigantic black bear roams around with his head hanging low.*

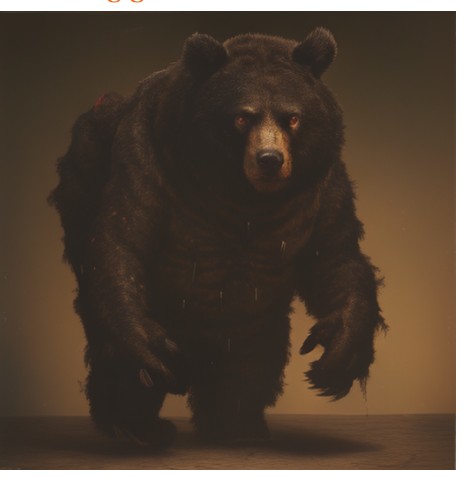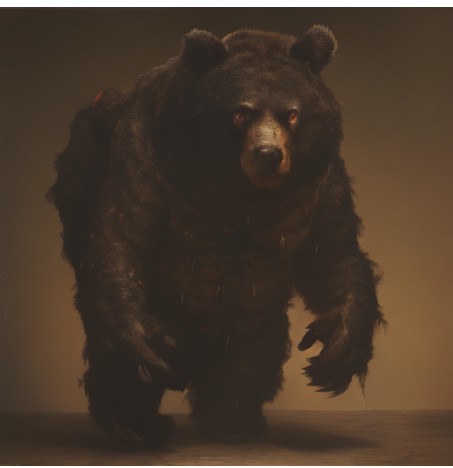

*A bunch of fruit sits in front of a portrait.*

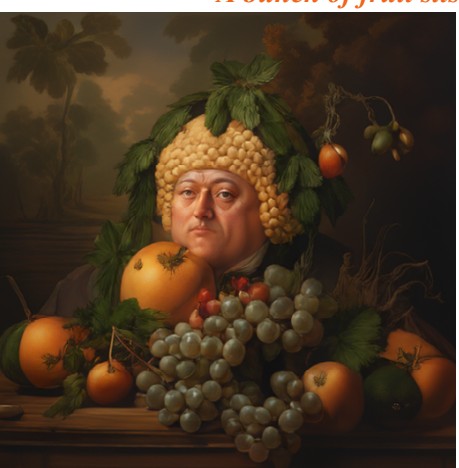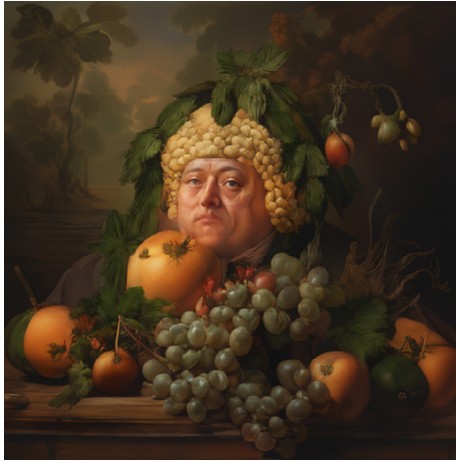

*A cat lying in the sun on a table.*

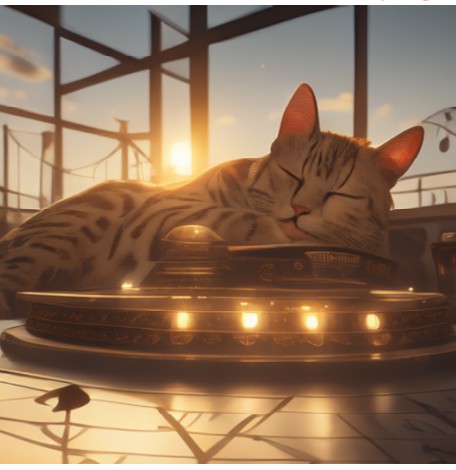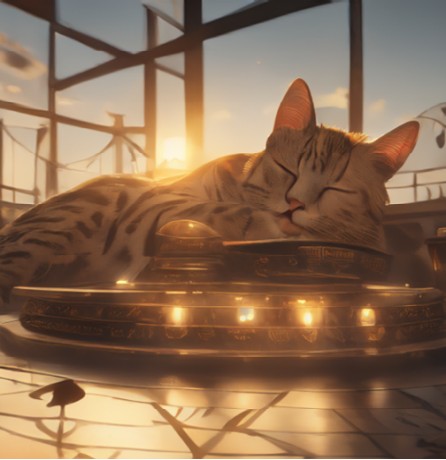

Figure 12: More qualitative comparison results between non-watermarked images and IndexMark watermarked images.

