# OpenReview forum: "Training-Free Watermarking for Autoregressive Image Generation"
_ICLR.cc/2026/Conference — ICLR 2026 Conference Withdrawn Submission_

### Official Review · Reviewer_Srdz · 2025-10-28

**Soundness:** 2
**Presentation:** 2
**Contribution:** 2
**Rating:** 2
**Confidence:** 4

**Summary:**

This paper proposes a watermarking framework named IndexMark, which is specially designed for autoregressive image generation model. Based on the redundancy of the codebook, this method divides the codebook index into red and green pairs and embeds the watermark based on the index replacement strategy guided by confidence in the generation process. Systematic experimental results demonstrate the effectiveness of the proposed method.

**Strengths:**

- The methodological design is effective.  Through the red-green index matching of codebook and the index replacement mechanism based on confidence guidance, the invisible embedding of watermark is realized.
- Adequate experiments and analysis demonstrate the effectiveness of the proposed method in terms of watermark verification accuracy.

**Weaknesses:**

There are some issues that need to be addressed:
- **Dependence of index encoder.** As shown in fig. 6(b), IndexMark strongly depends on the performance of index encoder. It is necessary to retrain such an encoder for each different VAR model, which is not the training-free watermarking scheme mentioned in the article title. And if there are 512×512 or 1024×1024 images and a larger number of codebook items, it is not sure whether the index encoder can be trained well. The author should provide or estimate the performance and efficiency of the index encoder and the training resources needed.
- **Redundancy of codebook.** The core premise of the method is that there are a large number of interchangeable similar vector pairs in the codebook. However, this redundancy may not be evenly distributed, and the key feature vectors may lack alternatives, so there is distortion in direct substitution. The effectiveness of the method is highly dependent on the quality of the codebook obtained by VQ-VAE training. In this paper, the robustness of the method under different codebooks is not tested.
- **Multi-user identification.** This paper mentions that identification can be provided according to the division of codebook. However, there is no experiment to illustrate this point in the experiment, and whether it can supplement the experiment of multi-user traceability detection. For example, if only one or a few pairing items are modified in the two codebook partitions, can the scheme distinguish users well? Or, at least modify the matching relationship of the unit index to distinguish?
- **Selection of threshold.** The threshold for the IndexMark method is selected based on unknown criteria. We noticed that the author used a different threshold strategy when facing the crop attack. If the threshold can be changed with the attack to improve the accuracy, how to determine the threshold when the attacker carries out unknown attacks (not common attacks)? We hope to see more experiments and discussions on the selection of threshold.
- **Natural generation detection.** When detecting the accuracy of watermark extraction, we guess that the author does not consider whether the naturally generated image (normal VAR generation) can be detected without watermark. It should provide more detailed experimental instructions and results.
- **Limited Baselines for Comparison.** The evaluation includes only outdated post-processing baselines all developed at least five years ago (2019 or earlier). To strengthen the comparison, the authors should incorporate more recent and robust methods, such as TrustMark [1] (2023) and VINE [2] (2025). Including these would provide a more thorough and up-to-date assessment of the proposed approach.
- **Image quality.** We observe that the PSNR value of this scheme is relatively low (usually more than 30 is considered as good), and it is not compared with the post-processing methods mentioned in the paper, such as DwtDct, DwtDctSvd and RivaGAN. Please supplement more relevant experiments and analysis.

Reference:

[1] TrustMark: Universal Watermarking for Arbitrary Resolution Images

[2] Robust Watermarking Using Generative Priors Against Image Editing: From Benchmarking to Advances

**Questions:**

See weaknesses. I would rate a weak rejection for this one (between 2: reject not good enough and 4: marginally below acceptance but wouldnt mind if the paper is accepted)

---

### Official Review · Reviewer_Ec1y · 2025-10-30

**Soundness:** 3
**Presentation:** 4
**Contribution:** 3
**Rating:** 6
**Confidence:** 3

**Summary:**

The paper considers the problem of watermarking images generated autoregressively.

In autoregressive image generation, we convert images to token representations in a latent space. The model is trained to predict the next token during training. Then, at inference time, the model starts with (I think) an empty vector (and I guess a prompt) and iteratively predicts the next token. Once the latent vector is filled in, we decode it to an image in pixel space.

This paper proposes a method of watermarking in the latent space: Each token is paired with a "nearby" neighbor token. And half the tokens are marked as "green" and half are marked as "red". Right before decoding the latent vector (and after it's been fully generated), the approach replaces some fraction of red tokens with their green pair.

It's important that the red and green tokens are close. In this work, the authors formulate a weighted perfect matching problem where each node is a token, and there are edges between the most related tokens as measured by cosine similarity. Then the blossoms algorithm is run to find a perfect matching.

When replacing tokens with their green versions, pairs with close relative probability (given by the autoregressive model) are prioritized. In order to deal with cropping, more ways of encoding the image are considered.

**Strengths:**

* Natural idea to use LLM red-green watermarking scheme for autoregressive images.

* Using the perfect matching algorithm as a subroutine to find "nearby" pairs is quite nice.

* The method performs well experimentally. (Although the number of existing watermarking schemes for *autoregressively* generated images is low.)

**Weaknesses:**

I like the paper a lot overall. There are questions below that I would like answered. But, even if all my questions are addressed, I'm worried about the technical contribution of this work. While well executed, the idea is to just (somewhat cleverly) apply red-green list watermarking to autoregressive images.

Another concern is that I'm not sure autoregressive image watermarking should be considered so differently from diffusion watermarking. It seems plausible to take a diffusion generated image vector in latent space, and apply the same red-green codebook strategy to watermark it. On one hand, this would increase the applicability of this work. On the other, I would like to then see comparison to all the SOTA diffusion watermarking methods (there are far more diffusion watermarking approaches than autoregressive watermarking approaches).

**Questions:**

1. When addressing cropped images, you effectively give each image more chances to get a higher green index rate. This changes the distribution, do you account for this when choosing the decision threshold? If not, how could you account for this?

2. Why are there some relative confidence values below 0? I would have thought the inference algorithm would choose the token with the *largest* probability, so $p_k \geq p_k'$ and $\log(p_k/p_k') \geq 0$.

3. Since it seems like the most technical contribution, I'd like to see evaluations of the perfect matching algorithm. Across many autoregressive token predictions of a particular token $t$, could you compare the average relative confidence of the chosen pair $t'$ vs the average relative confidence of the best token $t*$?

4. This is somewhat tangential, but you claim the LlamaGen model from 2024 is SOTA. I find this surprising given how fast-paced progress is in generative AI. Could you confirm?

---

### Official Review · Reviewer_oyKJ · 2025-10-31

**Soundness:** 3
**Presentation:** 3
**Contribution:** 2
**Rating:** 4
**Confidence:** 5

**Summary:**

This paper proposes IndexMark, a training-free watermarking framework for autoregressive (AR) image generation models. The method pairs metrically similar tokens and assigns them to red/green lists. During generation, it applies a "match-then-replace" strategy, swapping generated "red" indices with their corresponding "green" partners, guided by the model's prediction confidence. Watermark verification is done by calculating the proportion of "green" indices in the final image , a process that can be improved by an optionally trained "Index Encoder".

**Strengths:**

S1. Use case: The paper addresses the important and timely problem of watermarking autoregressive image models, which is a less-explored area than watermarking diffusion models.

S2. Clear Writing: The paper is well-written, and the proposed method is explained clearly.

S3. Strong core concepts: The method is built on severa ideas that I found very good: (a) leveraging codebook redundancy, (b) formalizing the index pairing as a maximum weight perfect matching problem to maximize intra-pair similarity, (c) the "match-then-replace" strategy, and (d) the optional Index Encoder to improve detection robustness.

S4. Good experimental scale: Experiments are conducted on state-of-the-art models (LlamaGen) at multiple resolutions (256x256, 384x384, 512x512) and on standard large-scale datasets (MSCOCO, ImageNet).

**Weaknesses:**

W1. Statistical Test: The watermark verification method relies on the CLT to approximate the distribution of the green index rate. Since the statistic follows a binomial distribution, for which an exact test or a more accurate confidence interval (e.g., see https://en.wikipedia.org/wiki/Binomial_proportion_confidence_interval#Clopper%E2%80%93Pearson_interval) could be used instead of a normal approximation.

W2. Weak post-hoc baselines: The post-hoc baselines (DwtDct, DwtDctSvd, RivaGAN)  are not state-of-the-art and are known to be weak. Their inclusion does not provide a meaningful comparison. More robust, state-of-the-art methods (e.g., TrustMark, Invismark, WAM) could be included.

W3. Misleading comparison to in-generation baselines: The paper argues that LLM watermarking methods (KGW, STA-1) "severely compromise" image quality. This claim is based on PSNR/SSIM metrics (Table 1), which are inappropriate for comparing generative methods that are not expected to produce the same image. The metrics that are appropriate (FID and CLIP) show that these baselines perform almost identically to the non-watermarked model (e.g., 25.81 FID vs. 25.53; 15.15 FID vs. 15.08), contradicting the paper's central argument. Besides I imagine that KGW and STA-1, because they generate images that are more different, are more robust against attacks like autoencoders or diffpure.

W4. Missing Baselines: The paper fails to cite or compare against several highly relevant and concurrent works on watermarking for AR image generation:
- A Watermark for Auto-Regressive Image Generation Models, https://arxiv.org/abs/2506.11371
- Radioactive Watermarks in Diffusion and Autoregressive Image Generative Models, https://arxiv.org/abs/2506.23731
- Watermarking Autoregressive Image Generation, https://arxiv.org/abs/2506.16349

W5. Incomplete robustness evaluation: The robustness evaluation in Table 2  is missing key geometric transformations. Token-based methods are notoriously vulnerable to attacks like crop-and-resize, horizontal flipping, and perspective changes, none of which are tested. The "Rot" (rotation) attack is only evaluated up to 0.5 degrees, which is not very meaningful in my opinion.

W6 (Minor). When discussing deep encoder-decoder methods, the paper cites RivaGAN rather than a more foundational work like HiDDeN (Zhu et al., 2018).

**Questions:**

Q1 (W3, W5): Could the authors clarify the claim of "severe quality degradation" for KGW/STA-1, given that their FID and CLIP scores in Table 1 are nearly identical to the original model? Furthermore, is it not likely that KGW/STA-1, being in-process, are more robust to common attacks (like diffusion purification) than the proposed post-generation "match-then-replace" method?

Q2 (W5): Why was the robustness to rotation limited to only 0.5 degrees? How does IndexMark perform against standard geometric attacks like random crop-and-resize (e.g., 50-90% area) or horizontal flipping?

---

### Official Review · Reviewer_2yed · 2025-11-12

**Soundness:** 2
**Presentation:** 2
**Contribution:** 2
**Rating:** 6
**Confidence:** 4

**Summary:**

The paper proposes IndexMark, a training-free watermarking framework designed for autoregressive image generation models (AR models). This framework aims to tackle the challenges of watermarking images generated by these models, which are often susceptible to misuse, such as in copyright issues or model manipulation. Unlike existing methods, which are designed for diffusion models, IndexMark specifically addresses the unique properties of AR models, embedding invisible watermarks using a match-then-replace strategy that minimally impacts image quality. By leveraging the redundancy in the codebook of AR models, the method replaces indices with similar tokens, ensuring imperceptibility while maintaining the diversity and quality of generated images. The paper also introduces a robust verification scheme, which includes an optional Index Encoder to enhance watermark detection accuracy and a cropping-resistant validation approach. Extensive experiments show that IndexMark achieves state-of-the-art performance, offering robustness against various image perturbations and attack scenarios.

**Strengths:**

Originality: The paper addresses a niche yet crucial gap in the field of image watermarking by focusing on autoregressive image generation models. This is a relatively underexplored area compared to watermarking techniques for diffusion models. The idea of using the redundancy in the codebook for embedding watermarks through a match-then-replace strategy is novel and provides an elegant solution to the imperceptibility problem.
Quality: The framework is well-constructed, with clear explanations of its methodology. The authors propose a training-free watermarking approach, which is a significant contribution, as it avoids the need for model retraining, making the watermarking process more practical and scalable. The use of the Blossom algorithm to solve the maximum weight perfect matching problem is a sound technical approach, and the detailed watermark verification system provides a robust mechanism for watermark detection.
Clarity: The paper is generally clear and well-structured. The methodology is thoroughly explained with appropriate mathematical formulations. The watermark embedding process, including the confidence-guided index replacement and statistical probability-based verification, is well-articulated. Diagrams and figures in the methodology section help illustrate the concepts effectively.
Significance: This work has significant implications for content moderation, copyright protection, and accountability in AI-generated content. As generative models like AR-based image generators become more prevalent, ensuring the traceability and ownership of generated images will become increasingly important, and IndexMark offers a practical solution.

**Weaknesses:**

Verification Performance at Varying Watermark Strengths: The paper briefly mentions watermark robustness under different conditions but lacks detailed analysis of verification performance across different watermark strengths. Specifically, the effect of varying watermark strength on verification accuracy (e.g., false positives/negatives) is not sufficiently explored. It would be beneficial to evaluate the performance at lower and higher confidence thresholds and under different attack types to better understand the trade-offs in verification accuracy.
Practical Scalability and Codebook Redundancy: While the method is grounded in leveraging redundancy in the codebook, the paper does not fully address the challenges or scalability when applying this approach to models with larger or more complex codebooks. A more detailed discussion on how the method scales, especially in cases where redundancy is less pronounced, would help assess its practical viability for real-world applications.

**Questions:**

Watermark Strength and Verification Accuracy: Could you provide a more detailed analysis of how the verification accuracy (false positives/negatives) changes with varying watermark strength and confidence thresholds? Specifically, how does performance vary across different attack conditions and levels of watermark intensity?

Scalability with Larger Codebooks: How does the proposed method perform when applied to autoregressive models with larger or more complex codebooks, where redundancy may be less pronounced? Are there any scalability issues or limitations that arise in such scenarios, and how might the method adapt to models with different codebook structures?

Impact on Image Diversity: Does embedding the watermark with the match-then-replace strategy affect the diversity of generated images? If so, how can this be mitigated while maintaining watermark robustness?

---

### Note · Authors · 2025-11-14

I have read and agree with the venue's withdrawal policy on behalf of myself and my co-authors.